# Clinical Characteristics and Risk Factors for Fatality and Severity in Patients with Coronavirus Disease in Korea: A Nationwide Population-Based Retrospective Study Using the Korean Health Insurance Review and Assessment Service (HIRA) Database

**DOI:** 10.3390/ijerph17228559

**Published:** 2020-11-18

**Authors:** Seung-Geun Lee, Geun U. Park, Yeo Rae Moon, Kihoon Sung

**Affiliations:** 1Division of Rheumatology, Department of Internal Medicine, Pusan National University School of Medicine, Pusan National University Hospital, Busan 49241, Korea; 2Biomedical Research Institute, Pusan National University Hospital, Busan 49241, Korea; 3LINEWALKS Inc., Seoul 06235, Korea; geunu@linewalks.com (G.U.P.); yeorae@linewalks.com (Y.R.M.); khsung@linewalks.com (K.S.)

**Keywords:** coronavirus, pandemic, comorbidity, death, prognosis

## Abstract

Background: We aimed to investigate the clinical characteristics and risk factors for fatality and severity in these patients. Methods: In this nationwide population-based retrospective study, we investigated the data of 7339 laboratory-confirmed COVID-19 patients, aged ≥ 18 years, using the Korean Health Insurance Review and Assessment Service (HIRA) database. Comorbidities and medications used were identified using HIRA codes, and severe COVID-19 was defined as that requiring oxygen therapy, mechanical ventilator, cardiopulmonary resuscitation, or extracorporeal membrane oxygenation. The outcomes were death due to COVID-19 and COVID-19 severity. Results: Mean patient age was 47.1 years; 2970 (40.1%) patients were male. Lopinavir/ritonavir, hydroxychloroquine, antibiotics, ribavirin, oseltamivir, and interferon were administered to 35.8%, 28.4%, 38.1%, 0.1%, 0.3%, and 0.9% of patients, respectively. After adjusting for confounding factors, diabetes mellitus, chronic kidney disease, previous history of pneumonia, aging, and male were significantly associated with increased risk of death and severe disease. No medication was associated with a reduced risk of fatality and disease severity. Conclusions: We found several risk factors for fatality and severity in COVID-19 patients. As the drugs currently used for COVID-19 treatment have not shown significant efficacy, all efforts should be made to develop effective therapeutic modalities for COVID-19.

## 1. Introduction

A novel coronavirus, severe acute respiratory syndrome coronavirus 2 (SARS-CoV-2, previously 2019-nCoV), was identified to have caused an outbreak in Wuhan, the capital city of Hubei province of China, in December 2019. The virus rapidly spread to the neighboring Asian countries and then worldwide. Although the initial outbreak appeared to have started as a zoonotic transmission from wild animals traded in a seafood market in Wuhan, it is evident that human-to-human transmission also occurs through droplets and close contact [1,2], and it is recognized as a major contributor to the spread of SARS-CoV-2 infection in most countries. SARS-CoV-2 infections are characterized by a wide spectrum of clinical manifestations, including asymptomatic infection, mid-upper respiratory illness, and severe viral pneumonia and was designated as coronavirus disease-19 (COVID-19) by the World Health Organization (WHO) in February 2020. Despite tremendous efforts worldwide, the WHO had to declare COVID-19 as a pandemic on 11 March 2020, because the global incidence of the disease increased sharply. The pandemic has not been controlled, as of 30 July 2020, because therapeutics or vaccines have not yet been developed. More than 16,812,755 confirmed cases, and 662,095 deaths have been reported globally [3]. Therefore, this pandemic is considered an emerging global health threat.

As the pandemic evolves, it is essential to understand its clinical characteristics and disease course. As fatal organ dysfunction such as shock, acute respiratory distress syndrome, acute kidney injury, and acute heart failure can occur in severe COVID-19 cases [4,5,6], it is necessary to identify prognostic factors and treatment strategies to improve clinical outcomes. Previous reports from many countries, including China [1,4,5,7], Italy [8,9], Iran [10], and the United States [11,12] have described the epidemiology, clinical features, and risk factors for poor prognosis in patients with COVID-19. However, most of these studies have been limited to a specific region of a country [5,8,11,12] or a single center [10]. In particular, the effects of current therapies such as antiviral agents and hydroxychloroquine on clinical outcomes have not been fully determined in population-based studies yet. Thus, we aimed to investigate the clinical characteristics and risk factors for fatality and severity in patients with COVID-19 using the Korean Health Insurance Review and Assessment Service (HIRA) database.

## 2. Materials and Methods

### 2.1. Study Design and Data Source

This population-based retrospective study was conducted to investigate confirmed cases of COVID-19 using the nationwide health insurance claim data (HIRA) released by the #OpenData4COVID19 project. On 26 March 2020, the Ministry of Health and Welfare (MOHW) of Korea shared the world’s first deidentified COVID-19 nationwide patient data from the HIRA database to help domestic and international researchers in finding measures to overcome the disease and provide evidence for effective policy enforcement [13]. The HIRA database consists of claims of approximately 98% of the Korean population (nearly 50 million individuals) and includes the beneficiary’s information such as age, sex, and address as well as data of the healthcare services availed, such as diagnosis, tests, prescriptions, and procedures. Specifically, the #OpenData4COVID19 project also provides health data, for the past 3 years, of those who underwent reverse transcription-polymerase chain reaction (RT-PCR) tests for COVID-19 in 2020 and regarding death. This study was approved by the Human Investigation Review Board of Public Institutional Bioethics Committee designated by the MOHW, which waived the requirement of informed consent due to the retrospective study design and anonymity of the HIRA database (IRB no. P01-202005-21-011).

### 2.2. Study Population and Assessment

In the HIRA database, all benefit claims relevant to COVID-19 were assigned a special “public crisis” code (MT043) for easy identification. In addition, a confirmed case could be identified using another HIRA code provided in the #OpenData4COVID19 project [13]. On the cutoff date, 15 May 2020, the data of 234,427 individuals who were tested for COVID-19 were identified, and the data of 7339 patients positive for COVID-19, aged ≥ 18 years, were evaluated (Figure 1). The primary outcome in this study was death due to COVID-19, which was identified according to a HIRA code. The secondary outcome was severe COVID-19, which was defined by the presence of claims for oxygen therapy, mechanical ventilator, cardiopulmonary resuscitation, or extracorporeal membrane oxygenation [14].

Data of the following variables of confirmed patients with COVID-19 were extracted from the HIRA database: Age, sex, residence, comorbidities, treatment duration, and medications used. According to their residence, the patients were categorized into 2 groups: Daegu city and Gyeongsangbuk-do province (DG), the largest epidemic region in Korea, and regions outside the DG area (Appendix A). Comorbidities potentially associated with death or severe COVID-19, such as influenza (KCD-7 J09-J11), tuberculosis (KCD-7 A15, A16, A19), chronic obstructive pulmonary disease (COPD; KCD-7 J431, J432, J438, J439, J44), pneumonia (KCD-7 J12-J18), asthma (KCD-7 J45, J46), diabetes mellitus (DM; KCD-7 E10-E14), chronic kidney disease (CKD; KCD-7 N18), chronic liver disease (KCD-7 K70-K77), hypertension (KCD-7 I10-I13, I15), cardiovascular/cerebrovascular diseases (CVDs; KCD-7 I61-I64, I20-I22, I24, I25), malignancies (KCD-7 C15-C22, C25, C43, C44, C460, C53, C54, C61, C64, C67, C840, D001, D002, D010-D015, D017, D019, D04, D06, D070, D075, D090), and human immunodeficiency virus infection (HIV infection; KCD-7 B20-B24) were identified. Patients with COVID-19 were considered to have a comorbidity if there was one or more reimbursement for KCD-7 codes of the above-mentioned diseases in the past 3 years before the confirmation of COVID-19. Treatment duration was calculated as the time interval from the date of the first reimbursement to that of the last reimbursement for COVID-19 and included both outpatient visits and hospital admissions. The data of the following medications for COVID-19 were retrieved using the Anatomical Therapeutic Chemical codes and HIRA general name codes: Lopinavir/ritonavir, hydroxychloroquine, ribavirin, type I interferon, human immunoglobulin G, oseltamivir, and antibiotics. Antibiotics were further categorized as penicillins, cephalosporins, quinolones, macrolides, aminoglycosides, sulfamethoxazole, trimethoprim, tetracyclines, and others (glycopeptides, oxazolidones, colistin, fosfomycin, and metronidazole). A detailed list of antibiotics is shown in Appendix A.

### 2.3. Statistical Analyses

Two-sided *p* < 0.05 was considered to indicate statistical significance, and all data analyses were carried out using SAS version 9.4 (SAS Inst., Cary, NC, USA). Data were presented as mean ± standard deviation (SD) for continuous variables and numbers with percentages for categorical variables. Group comparisons were conducted using a Student’s t-test for continuous variables and chi-square test or Fisher’s exact test for categorical variables, as appropriate. We hypothesized that certain comorbidities or medications were associated with the death or severity of COVID-19. To demonstrate this hypothesis, we performed 2 models of the multivariable logistic regression analysis. Multivariable model 1 included all variables regarding medications and comorbidities and variables with *p* < 0.05 in the univariable logistic regression analyses. Multivariable model 2 included one medication at a time with all variables regarding comorbidities and significant variables in the univariable analyses and one comorbidity at a time with all variables regarding medications and significant variables in the univariable analyses, thereby constructing individual models for each medication and comorbidity to minimize the possibility of overfitting multivariable model 1. Odds ratios (OR) and 95% confidence intervals (CI) were used to express significant associations. Owing to concerns about disclosure of personal information, the #OpenData4COVID19 project did not provide the date of death for patients with COVID-19, and thus, we were unable to estimate the Kaplan–Meier survival function and conduct Cox proportional hazards regression analysis.

## 3. Results

The data of the demographics and clinical characteristics of the 7339 patients included in this study are summarized in Table 1. The mean ± SD age of patients was 47.1 ± 19 years, and a peak prevalence of COVID-19 was observed in the age group of 20–29 years (25.2%), followed by those of 50–59 years (20.5%). A total of 2970 (40.1%) patients were male, and 4102 (55.9%) resided in the DG region. A total of 3038 (45.1%) patients had one or more comorbidities, and the most common comorbidity was hypertension (18.6%), followed by DM (11.6%), chronic liver diseases (8.7%), influenza (7.9%), pneumonia (6.9%), CVDs (6.1%), and asthma (5.4%). A total of 2820 (38.1%) patients had received at least one class of antibiotics for the treatment of COVID-19. The most common class of antibiotic used was quinolones (23.5%), cephalosporins (22.3%), macrolides (15.6%), and penicillins (8.7%). Lopinavir/ritonavir and hydroxychloroquine were administered to 35.8% and 28.4% of patients with COVID-19 infection, respectively.

A total of 227 (3.1%) deaths were reported as of 15 May 2020. Among the deceased, the proportions of males (53.3% vs. 40.1%, *p* < 0.001) and older patients (77.1 vs. 46.1 years, *p* < 0.001) were found to be higher than those of females and survivors (Table 1). Moreover, the proportion of deceased patients was significantly higher (88.5% vs. 55.4%, *p* < 0.001) than that of survivors in the DG region than that of patients residing in regions outside the DG province. A greater number of deceased patients had influenza, tuberculosis, COPD, pneumonia, asthma, DM, CKD, hypertension, CVDs, malignancies, and HIV infections (Table 1). Except for oseltamivir, the frequency of medications used for COVID-19 management was significantly higher in deceased patients than in survivors. Particularly, most deceased patients (92.1%) had received antibiotic therapy. In addition, more than 60% of the deceased patients had been treated with lopinavir/ritonavir or hydroxychloroquine.

A total of 927 (12.6%) patients were classified as severe cases. Similar to that of the deceased, patients with severe disease also had a significantly higher proportion of males and more commonly resided in the DG area than the non-severe counterparts (Table 1). The incidence of all comorbidities, except for HIV infection, was significantly higher in patients with severe COVID-19 than in non-severe patients. In addition, patients with severe COVID-19 were more likely to have received all types of medications for COVID-19 than non-severe counterparts. The frequency of death was also significantly higher in severe cases than in non-severe cases (21.6% vs. 0.4%, *p* < 0.001).

Results of the univariable and multivariable logistic regression models for death are shown in Table 2. In the univariable analyses, aging, male sex, DG area (a high epidemic region in Korea), tuberculosis, COPD, pneumonia, asthma, DM, CKD, hypertension, CVDs, malignancies, lopinavir/ritonavir, hydroxychloroquine, ribavirin, type I interferon, human immunoglobulin G, and antibiotics used were associated with a higher risk of death. In both the multivariable analyses models, comorbidities such as DM (OR 2.17, 95% CI 1.55–3.03; model 1 and OR 2.21, 95% CI 1.59–3.08; model 2) and CKD (OR 3.11, 95% CI 1.33–7.3; model 1 and OR 3.21, 95% CI 1.35–7.63; model 2) were found to be significant risk factors for death. Multivariable analysis model 1 revealed that a previous history of pneumonia was associated with a higher risk of death (OR 1.6, 95% CI 1.07–2.39). No medication that could reduce the risk of death due to COVID-19 was found in the multivariable logistic regression models, but rather, type I interferon (OR 4.76, 95% CI 2.48–9.11; model 1 and OR 6.54, 95% CI 3.52–12.14; model 2), human immunoglobulin G (OR 37.23, 95% CI 13.95–99.37; model 1 and OR 52.22, 95% CI 19.92–136.95; model 2), antibiotics (OR 4.18, 95% CI 2.36–7.42; model 1 and OR 4.41, 95% CI 2.65–7.35; model 2), and lopinavir/ritonavir (OR 1.68, 95% CI 1.22–2.32; model 2) were associated with increased risk of fatality in both the multivariable models. Hydroxychloroquine did not show any significant association with death in the multivariable models. The effect of individual classes of antibiotics on death was further analyzed, as presented in Appendix A. There was no class of antibiotics associated with improved survival. Rather, all classes of antibiotics, except tetracyclines, showed a significant association with higher fatality in multivariable models 1 and/or 2.

Associated factors for severe COVID-19, which were analyzed using logistic regression models, are listed in Table 3. Comorbidities such as pneumonia, DM, and CKD were significantly associated with an increased risk of severe COVID-19 in both the multivariable models. All medications, except ribavirin, had a significantly higher probability of severe cases in both the multivariable models. The relationship between individual classes of antibiotics and severe COVID-19 is shown in Appendix A. All classes of antibiotics showed an increased risk of severe COVID-19 in multivariable models 1 and/or 2.

## 4. Discussion

This study analyzed the demographic and clinical features of 7339 adult Korean patients with COVID-19 and identified factors associated with fatality and severity of the disease using the nationwide health insurance database. Approximately 45.1% of the confirmed cases had at least one comorbidity. The crude case fatality rate (CFR) and the frequency of severe cases were reported to be 3.1% and 12.6%, respectively. A previous history of pneumonia, DM, and CKD was a significant risk factor for both the fatality and severity of COVID-19. In addition, significant effects of aging and predominance of males on death and disease severity were observed. However, no medications that could improve clinical outcomes were found in our analysis, suggesting an urgent need for the development of effective therapeutics for COVID-19.

After the first cases of COVID-19 were reported in China, numerous studies conducted worldwide have reported the clinical characteristics and outcomes of COVID-19. Although the disease can occur in individuals of any age, individuals of middle ages were reported to be most affected in the early reports from Western countries [15]. In contrast, our data showed that the age group of 20–29 years was most susceptible to COVID-19 in Korea. This discrepancy may be a result of the differences in sociocultural factors and quarantine norms of countries. For example, people in Western countries can be more exposed to close physical contact such as a hug and kiss as compared with those in Asian and Middle Eastern countries. In addition, there are significant differences in the degree of the restriction of social contact according to the countries. The crude CFR of COVID-19 considerably varied between 0.08% and 9.51% globally [16]. The highest crude CFR was reported in Italy (9.51%), followed by Iran (7.86%) and Spain (6.59%) [16]. Our data showed that the crude CFR of COVID-19 was 3.1% in Korea, which is significantly lower than that in Italy, Iran, and Spain. However, the crude CFR was calculated based on cross-sectional data; thus, it may be biased by the difference in the timespan of COVID-19 in the countries during the course of the ongoing pandemic, as pointed out by Kim et al. [16] As older patients are known to be susceptible to death due to COVID-19 [4,7,17,18], the difference in the aging structure of countries should be considered when interpreting CFR worldwide [16].

Underlying comorbidities may have significant effects on the occurrence, severity, and fatality of COVID-19. In our data, hypertension was the most common underlying disease in Korean COVID-19 patients, in accordance with previous meta-analyses of patients from other countries [19,20]. Hypertension is one of the most common chronic diseases affecting more than 30% of the adult population worldwide [21], which could explain this finding. However, no clear evidence has been found to demonstrate whether hypertension is an independent risk factor for SARS-CoV-2 infection [22], and this topic is beyond the scope of our study because we analyzed only COVID-19 patients and could not compare the frequency of certain comorbidity between confirmed cases and controls. In line with other studies [14,23,24,25], our data also revealed that individuals with DM were at a higher risk of COVID-19 fatality and severity. In previous studies, DM was linked to severe complications of COVID-19, such as multi-organ failure and acute respiratory distress syndrome, making the patients more vulnerable to various infectious diseases [23,24]. The deleterious effect of DM on COVID-19 may be due to its multifactorial nature, pro-inflammatory/pro-coagulative status, and various complications [24,26]. In addition, CKD was also found to have a strong association with fatality and severity of COVID-19 infection in this study. Likewise, previous studies also reported that patients with CKD showed a 3-fold higher risk for severe COVID-19 and a 12-fold higher frequency of intensive care unit admission [27]. CKD induces marked alterations in the immune system, leading to a state of chronic inflammation and immunosuppression [27], predisposing to the morbidity and fatality of COVID-19. In addition to these points, these comorbidities may play a role in making the decision for the mechanical ventilator, cardiopulmonary resuscitation, or extracorporeal membrane oxygenation. It is possible that patients with underlying diseases have received more intensive care due to concerns that they will be more susceptible to corona infection. According to our literature review, two studies investigated the association of comorbidities with the fatality and severity of COVID-19 using the HIRA database. However, these previous studies did not adjust the effect of treatment for SARS-CoV-2; thus, we presume that our study provides a comprehensive understanding of the effect of underlying comorbidities on the clinical outcomes of COVID-19.

To the best of our knowledge, this is the first study investigating the relationship between prescribed medications for COVID-19 and its prognosis using the nationwide HIRA data. However, we did not observe any drugs currently being used to treat COVID-19 in Korea, to have beneficial effects on survival and disease severity. Rather, antibiotics, lopinavir/ritonavir type I interferon, and human immunoglobulin G were associated with an increased risk of mortality in our data. Owing to the retrospective study design, the possibility of reverse causality should be considered when interpreting this finding. Prescribed medications for COVID-19 may be considered as a confounding bias for more severe patients in our study. Globally, specific and effective therapeutic modalities for COVID-19 have not been established yet [28]. Although chloroquine, lopinavir/ritonavir, ribavirin, and remdesivir have shown successful inhibition of SARS-CoV-2 in vitro, these drugs may not be effective in COVID-19 patients with persistent viremia [29]. Therefore, all efforts should be made to develop novel medications to prevent and treat SARS-CoV-2 infection effectively.

The DG area was associated with higher mortality in our data. Because the healthcare system, accessibility to medical services, and other clinical factors do not differ between DG and non-DG areas in Korea, these factors may not affect the higher mortality in the DG area. About 70% of the confirmed cases in the DG area are linked to religious groups of Shincheonji church located in Daegu, and thus, many confirmed cases have occurred explosively in the DG area, unlike the non-DG area. We assumed that these situations might affect the higher mortality rate in the DG area, although the exact potential causes for results are still unclear. Accordingly, further studies are needed to clarify the reasons for the high mortality rate of COVID-19 in the DG area.

The potential limitations of the current study need to be addressed. First, we investigated only laboratory-confirmed COVID-19 patients. As RT-PCR tests for COVID-19 were performed for those who were symptomatic or had a history of contact with confirmed cases in Korea, most of the asymptomatic patients may not be detected using data from the HIRA database and thus were not included in our study, which causes a selection bias. In addition, because this study did not include control subjects with negative results for RT-PCR, comparisons of demographic and clinical variables between confirmed cases and negative controls could not be conducted. Second, the HIRA database does not provide information regarding the results of blood or imaging tests and hemodynamic status, reflecting the severity of COVID-19, which could not be fully adjusted in our study. Third, because we defined each comorbidity based on diagnostic codes in the HIRA database, the severity of comorbidities could not be considered in this study. Lastly, as mentioned, the results could be potentially confounded by the retrospective nature of the study design.

## 5. Conclusions

In summary, the current study found that aging, male sex, DM, CKD, and previous history of pneumonia are significant risk factors associated with the fatality and severity of COVID-19. Thus, patients confirmed with COVID-19 and having these features can be considered as high-risk groups and require early identification, intensive monitoring, and aggressive management. The drugs currently used to treat COVID-19 have not shown significant efficacy; hence, all efforts should be made to develop effective therapeutic modalities for COVID-19. We believe that the nationwide HIRA data of COVID-19 confirmed cases, through the #OpenData4COVID19 project, would enhance the understanding of the epidemiological features of this disease, thus contributing to optimal patient care and management.

## Figures and Tables

**Figure 1 ijerph-17-08559-f001:**
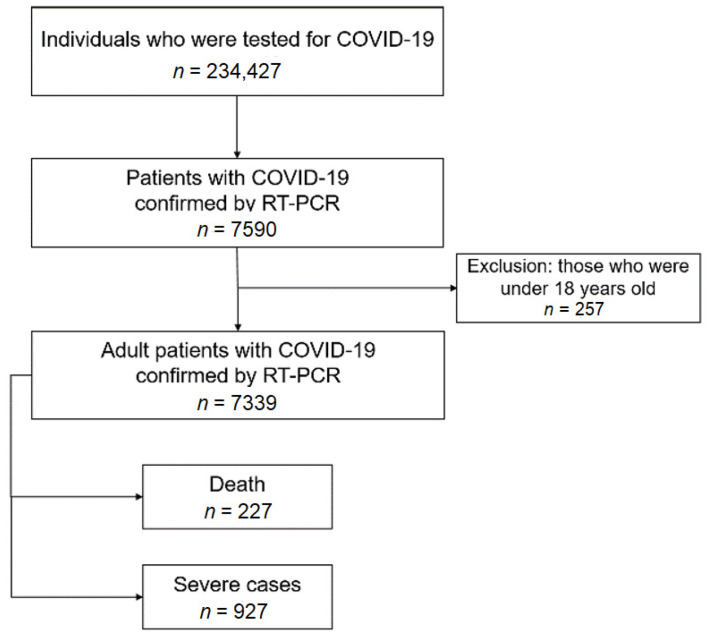
Flow chart of the study population.

**Table 1 ijerph-17-08559-t001:** Demographic and clinical characteristics of 7339 patients with coronavirus disease-19 infection.

	Total (*n* = 7339)	Deceased (*n* = 227)	Survivors (*n* = 7112)	*p* Value *	Severe Cases (*n* = 927)	Non-Severe Cases (*n* = 6412)	*p* Value ^§^
Age, years, mean ± SD	47.1 ± 19.0	77.1 ± 10.8	46.1 ± 18.4	<0.001	66.8 ± 15.2	44.2 ± 17.8	<0.001
Age group							
18–19 years, *n* (%)	180 (2.5)	0 (0)	180 (2.5)	<0.001	2 (0.2)	178 (2.8)	<0.001
20–29 years, *n* (%)	1852 (25.2)	0 (0)	1852 (26.0)		23 (2.5)	1829 (28.5)	
30–39 years, *n* (%)	777 (10.6)	2 (0.9)	775 (10.9)		28 (3.0)	749 (11.7)	
40–49 years, *n* (%)	1008 (13.7)	1 (0.4)	1007 (14.2)		46 (5.0)	962 (15.0)	
50–59 years, *n* (%)	1502 (20.5)	15 (6.6)	1487 (20.9)		173 (18.7)	1329 (20.7)	
60–69 years, *n* (%)	1056 (14.4)	35 (15.4)	1021 (14.4)		229 (24.7)	827 (12.9)	
70~ years, *n* (%)	964 (13.1)	174 (76.7)	790 (11.1)		426 (46)	538 (8.4)	
Male, *n* (%)	2970 (40.1)	121 (53.3)	2849 (40.1)	<0.001	441 (47.6)	2529 (39.4)	<0.001
Treatment duration, days, mean ± SD	22.0 ± 14.5	19.1 ± 15.9	22.1 ± 14.5	0.005	29.9 ± 18.7	20.9 ± 13.5	<0.001
DG area, *n* (%)	4139 (55.9)	201 (88.5)	3938 (55.4)	<0.001	705 (76.1)	3434 (53.6)	<0.001
Comorbidity							
Influenza, *n* (%)	582 (7.9)	10 (4.4)	572 (8.0)	0.046	49 (5.3)	533 (8.3)	<0.001
Tuberculosis, *n* (%)	28 (0.4)	4 (1.8)	24 (0.3)	<0.001	11 (1.2)	17 (0.3)	<0.001
COPD, *n* (%)	81 (1.1)	14 (6.2)	67 (0.9)	<0.001	32 (3.5)	49 (0.8)	<0.001
Pneumonia, *n* (%)	513 (6.9)	56 (24.7)	457 (6.4)	<0.001	173 (18.7)	340 (5.3)	<0.001
Asthma, *n* (%)	397 (5.4)	21 (9.3)	376 (5.3)	0.009	93 (10.0)	304 (4.7)	<0.001
DM, *n* (%)	857 (11.6)	91 (40.1)	766 (10.8)	<0.001	264 (28.5)	593 (9.2)	<0.001
CKD, *n* (%)	48 (0.6)	12 (5.3)	36 (0.5)	<0.001	28 (3.0)	20 (0.3)	<0.001
Chronic liver disease, *n* (%)	645 (8.7)	20 (8.8)	625 (8.8)	0.991	98 (10.6)	547 (8.5)	0.040
Hypertension, *n* (%)	1373 (18.6)	106 (46.7)	1267 (17.8)	<0.001	371 (40.0)	1002 (15.6)	<0.001
CVDs, *n* (%)	455 (6.1)	55 (24.2)	400 (5.6)	<0.001	155 (16.7)	300 (4.7)	<0.001
Malignancies, *n* (%)	162 (2.2)	20 (8.8)	142 (2.0)	<0.001	60 (6.5)	102 (1.6)	<0.001
HIV infection, *n* (%)	4 (0.1)	1 (0.4)	3 (0.0)	0.011	1 (0.1)	3 (0.0)	0.456
Medications							
Lopinavir/ritonavir, *n* (%)	2648 (35.8)	158 (69.6)	2490 (35.0)	<0.001	733 (79.1)	1915 (29.9)	<0.001
Hydroxychloroquine, *n* (%)	2102 (28.4)	141 (62.1)	1961 (27.6)	<0.001	574 (61.9)	1528 (23.8)	<0.001
Ribavirin, *n* (%)	4 (0.1)	2 (0.9)	2 (0.0)	<0.001	4 (0.4)	0 (-)	<0.001
Type I interferon, *n* (%)	63 (0.9)	28 (12.3)	35 (0.5)	<0.001	47 (5.1)	16 (0.2)	<0.001
Human immunoglobulin G, *n* (%)	36 (0.5)	24 (10.6)	12 (0.2)	<0.001	35 (3.8)	1 (0.0)	<0.001
Oseltamivir, *n* (%)	19 (0.3)	1 (0.4)	18 (0.3)	0.584	8 (0.9)	11 (0.2)	<0.001
Antibiotics, *n* (%)	2820 (38.1)	209 (92.1)	2611 (36.7)	<0.001	844 (91.0)	1976 (30.8)	<0.001
Penicillins, *n* (%)	646 (8.7)	149 (65.6)	497 (7.0)	<0.001	404 (43.6)	242 (3.8)	<0.001
Cephlosporins, *n* (%)	1649 (22.3)	156 (68.7)	1493 (21.0)	<0.001	611 (65.9)	1038 (16.2)	<0.001
Quinolones, *n* (%)	1739 (23.5)	159 (70.0)	1580 (22.2)	<0.001	603 (65.0)	1136 (17.7)	<0.001
Macrolides, *n* (%)	1152 (15.6)	103 (45.4)	1049 (14.7)	<0.001	407 (43.9)	745 (11.6)	<0.001
Aminoglycosides, *n* (%)	45 (0.6)	8 (3.5)	37 (0.5)	<0.001	28 (3.0)	17 (0.3)	<0.001
TMP/SMX, *n* (%)	43 (0.6)	8 (3.5)	35 (0.5)	<0.001	34 (3.7)	9 (0.1)	<0.001
Tetracyclines, *n* (%)	33 (0.4)	5 (2.2)	28 (0.4)	<0.001	16 (1.7)	17 (0.3)	<0.001
Others, *n* (%)	238 (3.2)	89 (39.2)	149 (2.1)	<0.001	195 (21.0)	43 (0.7)	<0.001

* Indicates comparison between deceased and survivors. ^§^ Indicates comparison between severe and non-severe cases. SD: Standard deviation. DG: Daegu city and Gyeongsangbuk-do province. COPD: Chronic obstructive pulmonary disease. DM: Diabetes mellitus. CKD: Chronic kidney disease. CVDs: cardiovascular/cerebrovascular diseases. HIV: Human immunodeficiency virus. TMP/SMX: Trimethoprim and sulfamethoxazole.

**Table 2 ijerph-17-08559-t002:** Results of multivariable logistic models for death in patients with coronavirus disease-19 infection.

	Univariable Model		Multivariable Model 1 *		Multivariable Model 2 ^§^	
	OR (95% CI)	*p* Value	OR (95% CI)	*p* Value	OR (95% CI)	*p* Value
Influenza	0.53 (0.28–1.00)	0.050	0.92 (0.42–2.03)	0.837	1.03 (0.49–2.18)	0.931
Tuberculosis	5.30 (1.82–15.40)	0.002	2.34 (0.61–8.94)	0.216	2.79 (0.71–10.89)	0.140
COPD	6.91 (3.82–12.49)	<0.001	1.39 (0.65–2.97)	0.397	1.34 (0.66–2.73)	0.422
Pneumonia	4.77 (3.48–6.54)	<0.001	1.6 (1.07–2.39)	0.024	1.47 (0.99–2.18)	0.055
Asthma	1.83 (0.15–2.90)	0.010	0.68 (0.38–1.21)	0.190	0.83 (0.48–1.43)	0.494
DM	5.54 (4.21–7.30)	<0.001	2.17 (1.55–3.03)	<0.001	2.21 (1.59–3.08)	<0.001
CKD	10.97 (5.63–21.39)	<0.001	3.11 (1.33–7.3)	0.009	3.21 (1.35–7.63)	0.008
Chronic liver disease	1.00 (0.63–1.60)	0.991	0.87 (0.50–1.51)	0.609	1.01 (0.58–1.74)	0.976
Hypertension	4.04 (3.09–5.28)	<0.001	0.85 (0.61–1.18)	0.323	0.76 (0.55–1.06)	0.102
CVDs	5.37 (3.90–7.39)	<0.001	0.82 (0.56–1.22)	0.330	0.95 (0.64–1.40)	0.778
Malignancies	4.74 (2.91–7.73)	<0.001	1.04 (0.56–1.93)	0.895	1.21 (0.67–2.18)	0.535
HIV infection	5.54 (4.21–7.30)	<0.001	106.93 (6.38–>999)	0.001	101.92 (5.92–>999)	0.001
Lopinavir/ritonavir	4.25 (3.19–5.66)	<0.001	0.81 (0.56–1.18)	0.268	1.68 (1.22–2.32)	0.002
Hydroxychloroquine	4.31 (3.28–5.66)	<0.001	0.86 (0.61–1.21)	0.380	1.22 (0.89–1.68)	0.215
Ribavirin	31.54 (4.42–224.93)	<0.001	5.82 (0.34–99.64)	0.225	>999 (410.48–>999)	0.001
Type I interferon	28.45 (16.97–47.69)	<0.001	4.76 (2.48–9.11)	<0.001	6.54 (3.52–12.14)	<0.001
Human immunoglobulin G	69.95 (34.50–141.83)	<0.001	37.23 (13.95–99.37)	<0.001	52.22 (19.92–136.95)	<0.001
Oseltamivir	1.74 (0.23–13.12)	0.589	3.74 (0.45–31.04)	0.221	2.98 (0.34–26.11)	0.324
Antibiotics	20.02 (12.34–32.47)	<0.001	4.18 (2.36–7.42)	<0.001	4.41 (2.65–7.35)	<0.001
Age, years	1.13 (1.11–1.14)	<0.001	1.12 (1.11–1.14)	<0.001	-	-
Male	1.71 (1.31–2.23)	<0.001	2.10 (1.51–2.92)	<0.001	-	-
DG area	6.23 (4.13–9.40)	<0.001	2.06 (1.27–3.34)	0.003	-	-

* Included all variables regarding medications and comorbidities and variables with *p* < 0.05 in the univariable logistic regression analyses. ^§^ Included one medication at a time with all variables regarding comorbidities and significant variables in the univariable analyses and one comorbidity at a time with all variables regarding medications and significant variables in the univariable analyses. COPD: Chronic obstructive pulmonary disease. DM: Diabetes mellitus. CKD: Chronic kidney disease. CVDs: Cardiovascular/cerebrovascular diseases. HIV: Human immunodeficiency virus. DG: Daegu city and Gyeongsangbuk-do province.

**Table 3 ijerph-17-08559-t003:** Results of multivariable logistic models for severe coronavirus disease-19 infection.

	Univariable Model		Multivariable Model 1 *		Multivariable Model 2 ^§^	
	OR (95% CI)	*p* Value	OR (95% CI)	*p* Value	OR (95% CI)	*p* Value
Influenza	0.62 (0.46–0.83)	0.002	0.92 (0.64–1.32)	0.653	0.96 (0.67–1.37)	0.815
Tuberculosis	4.52 (2.11–9.68)	<0.001	2.23 (0.85–5.84)	0.103	3.06 (1.18–7.94)	0.021
COPD	4.64 (2.96–7.29)	<0.001	1.02 (0.57–1.82)	0.960	1.16 (0.67–2.00)	0.605
Pneumonia	4.10 (3.36–5.00)	<0.001	1.74 (1.34–2.24)	<0.001	1.79 (1.39–2.30)	<0.001
Asthma	2.24 (1.76–2.86)	<0.001	1.22 (0.89–1.68)	0.223	1.40 (1.03–1.91)	0.030
DM	3.91 (3.31–4.61)	<0.001	1.36 (1.11–1.68)	0.003	1.42 (1.16–1.74)	0.001
CKD	9.95 (5.58–17.74)	<0.001	3.40 (1.67–6.92)	0.001	3.55 (1.75–7.18)	<0.001
Chronic liver disease	1.27 (1.01–1.59)	0.041	0.86 (0.66–1.13)	0.291	0.97 (0.74–1.27)	0.837
Hypertension	3.60 (3.11–4.61)	<0.001	1.05 (0.87–1.27)	0.601	0.99 (0.82–1.20)	0.933
CVDs	4.09 (3.32–5.04)	<0.001	0.98 (0.75–1.27)	0.847	1.09 (0.84–1.40)	0.514
Malignancies	4.28 (3.09–5.94)	<0.001	1.26 (0.83–1.91)	0.271	1.30 (0.86–1.95)	0.208
HIV infection	2.31 (0.24–22.20)	0.469	5.92 (0.43–81.22)	0.183	8.60 (0.71–103.83)	0.091
Lopinavir/ritonavir	8.87 (7.51–10.49)	<0.001	2.47 (2.02–3.03)	<0.001	5.25 (4.38–6.29)	<0.001
Hydroxychloroquine	5.20 (4.50–6.01)	<0.001	1.64 (1.37–1.96)	<0.001	2.55 (2.16–3.02)	<0.001
Ribavirin	>999 (0–>999)	0.945	>999 (0–>999)	0.970	>999 (0–>999)	0.966
Type I interferon	21.35 (12.05–37.82)	<0.001	3.00 (1.59–5.65)	0.001	5.99 (3.16–11.37)	<0.001
Human immunoglobulin G	251.52 (34.42–>999)	<0.001	51.48 (6.76–392.23)	<0.001	146.16 (18.72–>999)	<0.001
Oseltamivir	5.07 (2.03–12.64)	<0.001	3.62 (1.26–10.41)	0.017	7.00 (2.39–20.50)	<0.001
Antibiotics	22.83 (18.11–28.78)	<0.001	5.63 (4.30–7.37)	<0.001	10.87 (8.52–13.88)	<0.001
Age, years	1.08 (1.07–1.08)	<0.001	1.05 (1.04–1.06)	<0.001	-	-
Male	1.39 (1.21–1.60)	<0.001	1.60 (1.34–1.91)	<0.001	-	-
DG area	2.75 (2.35–3.23)	<0.001	0.94 (0.77–1.15)	0.573	-	-

* Included all variables regarding medications and comorbidities and variables with *p* < 0.05 in the univariable logistic regression analyses. ^§^ Included one medication at a time with all variables regarding comorbidities and significant variables in the univariable analyses and one comorbidity at a time with all variables regarding medications and significant variables in the univariable analyses. COPD: Chronic obstructive pulmonary disease. DM: Diabetes mellitus. CKD: Chronic kidney disease. CVDs: Cardiovascular/cerebrovascular diseases. HIV: Human immunodeficiency virus. DG: Daegu city and Gyeongsangbuk-do province.

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
