# Peer review of "Clinical Characteristics and Risk Factors for Fatality and Severity in Patients with Coronavirus Disease in Korea: A Nationwide Population-Based Retrospective Study Using the Korean Health Insurance Review and Assessment Service (HIRA) Database"

_ijerph, 2020, doi:10.3390/ijerph17228559_

Round 1

Reviewer 1 Report

General comments:

The authors described the real-world evidence of clinical characteristics and risk factors for fatality and severity in patients with coronavirus disease using the Korean HIRA database.

This view angle is quite unconventional in the medical and health of COVID-19 infection.

Although many clinical trials are proceeding with try and error around the world, we need/seek universal therapeutic measures not only in developed countries but also in developing countries. The article will be suited to our requirements.

Minor comments:

Page. 4, Line165-166: Because the CI is a wide range, HIV infection is not a significant risk factor.

Table 3:  There is not an asterisk (*) and a section mar (§) in the multivariable model 1/2.

Discussion Page. 2/12. Line260-264: It is better to add the new reference, i.e. October 29, 2020. N Engl J Med 2020; 383:1724-1734 DOI: 10.1056/NEJMoa2026116

Author Response

Minor comments:

Page. 4, Line165-166: Because the CI is a wide range, HIV infection is not a significant risk factor.

: Thank you for your kind comment. We amended this sentence, as you suggested (highlighted in yellow).

Table 3:  There is not an asterisk (*) and a section mar (§) in the multivariable model 1/2.

: We added asterisk and mar in Table 3 (highlighted in yellow).

Discussion Page. 2/12. Line260-264: It is better to add the new reference, i.e. October 29, 2020. N Engl J Med 2020; 383:1724-1734 DOI: 10.1056/NEJMoa2026116

: We added the new reference in the discussion section, as you suggested.

Reviewer 2 Report

1) In the introduction section increase the information about risk factor to present severe covid19 disease. This topic is theme is underdevelopment 

2) Which is the prevalence of HTA, DM, chronic liver disease, CVDs and asthma in Korea? Please compare the prevalence of those disease with the reported in your study. 

3) Provide more information why at the DG region they have a higher mortality rate. The explanation is only is the high incidence in DG? or for example de prevalence of other co-morbidities (like DM, HTA, etc) are higher in DG region? or if the health system in DG region is different from the health system of the other regions?

4) How the authors can explain that in the death patients the use of antibiotic was higher than in survivors? They need more antibiotics due to more frequency of co-infections? or due the high severity of the disease without an specific treatment? or the excessive use of antibiotics increase the complication rate in covid19 patients?

5) One of the most important results of you paper is that the employ of medicaments did not reduce the risk of death. Therefore which will be your proposal about the use of antibiotics, interferon, antiviral, etc. Please, discuss more this result and make it more evident in the conclusions. 

Author Response

1) In the introduction section increase the information about risk factor to present severe covid19 disease. This topic is theme is underdevelopment

: Thank you for your kind comment. Due to limitations in health insurance claims data, we have not been able to analyze all of the medical factors related to the risk of severe covid19.

2) Which is the prevalence of HTA, DM, chronic liver disease, CVDs and asthma in Korea? Please compare the prevalence of those disease with the reported in your study.

: The prevalence of HTN, DM, chronic liver disease, CVDs and asthma in Korea is 22.1%, 13.2%, 13.6%, 6.4%, 0.8% based on HIRA database of 2018. The prevalence of HTN (p<0.001), DM (p<0.001), chronic liver disease (p<0.001), asthma (p<0.001) in confirmed cases of our study was significantly lower than those in general population. However, the comparisons of the frequency of comorbidities between confirmed cases and general population in Korea are beyond of scope in this study. In addition, because the #OpenData4COVID19 project which is used in our study do not provide the information of general population, the direct comparisons between confirmed cases and general population are not possible. Please consider this situation.

3) Provide more information why at the DG region they have a higher mortality rate. The explanation is only is the high incidence in DG? or for example de prevalence of other co-morbidities (like DM, HTA, etc) are higher in DG region? or if the health system in DG region is different from the health system of the other regions?

: Healthcare system, accessibility to medical services and other clinical factors do not differ between DG and non-DG area. Thus, these factors may not affect the higher mortality in DG area. About 70% of the confirmed cases in DG area are linked to a religious groups of Shincheonji church located in Daegu, and thus, many confirmed cases have occurred explosively in DG area, unlike non-DG area. We assumed that these situations might affect the higher mortality rate in DG area, although the exact potential causes for results are still unclear. Accordingly, further studies are needed to clarify the reasons for the high mortality rate of COVID-19 in DG area. We added this notion in the discussion section (highlighted in yellow).

4) How the authors can explain that in the death patients the use of antibiotic was higher than in survivors? They need more antibiotics due to more frequency of co-infections? or due the high severity of the disease without an specific treatment? or the excessive use of antibiotics increase the complication rate in covid19 patients?

: Owing to the retrospective study design, it is difficult to determine the exact reasons for the association between the antibiotics use and increased mortality. Antibiotics use in our data may be considered as a confounding bias for more severe patients. We added this notion in the discussion section (highlighted in yellow).  

5) One of the most important results of you paper is that the employ of medicaments did not reduce the risk of death. Therefore which will be your proposal about the use of antibiotics, interferon, antiviral, etc. Please, discuss more this result and make it more evident in the conclusions.

: As mentioned in the reply regarding comment 4, the employ of medicaments did not reduce the risk of death but rather increased the mortality rate. We conjecture that prescribed medications for COVID-19 may be like a confounding bias for more severe patients, although the exact reasons for this finding are still unclear. We added this notion in the discussion section (highlighted in yellow).  

Reviewer 3 Report

While this is a large population-based study which may contribute to the current literature, the retrospective nature and methodology of this study have not contributed to further new insight into the disease as the conclusion of the study has been described previously. However,  as this disease is new, there is merit to the study as a ‘snapshot’ of the pandemic in Korea.  

Minor changes

Line 42 – February 2019 should be February 2020

Major

  1. Discussion around the difference between DG vs non DG regions
  2. Are tuberculosis patients in these cohort active tuberculosis? What is the prevalence of TB and are TB analysis routine and were there increased TB incidence during the pandemic?
  3. How was bacterial pneumonia (presuming Covid-19 pneumonia in severe stages can present clinically like bacterial pneumonia) diagnosed? Radiologically? Microbiologically? This is a limitation of population-based study. It is likely not likely useful for ‘pneumonia’ to be included in the analyses as delineation is difficult (unlike COPD, hypertension, diabetes). I would suggest removing this altogether.
  4. It is also important to discuss around how these chronic and acute co-morbidities play in making the decision for further escalation such as ICU admission, intubation and ventilation as well advanced resuscitation care.
  5. Please discuss further Line 218 re: sociocultural factors and quarantine norm of countries more explicitly.
  6. For paragraph starting from Line 251, it’s important to emphasize that prescribed medications for Covid-19 is like a confounding bias for more severe patients. Although both uni and multivariable analyses were used, hospitalization may be a useful variable in the logistic regression analyses to reflect severe patients who were hospitalized and likely to be commenced on anti-viral, antibiotics etc.
  7. Also worth mentioning if systemic steroids and/or remdesivir data available due to recent insight on the benefits

Author Response

Line 42 – February 2019 should be February 2020

: We amended this sentence as you recommended (highlighted in yellow).

Major

1. Discussion around the difference between DG vs non DG regions

: Healthcare system, accessibility to medical services and other clinical factors do not differ between DG and non-DG area. Thus, these factors may not affect the higher mortality in DG area. About 70% of the confirmed cases in DG area are linked to a religious groups of Shincheonji church located in Daegu, and thus, many confirmed cases have occurred explosively in DG area, unlike non-DG area. We assumed that these situations might affect the higher mortality rate in DG area, although the exact potential causes for results are still unclear. Accordingly, further studies are needed to clarify the reasons for the high mortality rate of COVID-19 in DG area. We added this notion in the discussion section (highlighted in yellow).

2. Are tuberculosis patients in these cohort active tuberculosis? What is the prevalence of TB and are TB analysis routine and were there increased TB incidence during the pandemic?

: “Tuberculosis” in these cohort indicates the previous history of active TB, as we had described in the method section. It means that the diagnosis of active TB before the occurrence of COVID-19 confirmation. In this study, TB incidence was not analyzed during the pandemic of COVID-19 in this study.     

3. How was bacterial pneumonia (presuming Covid-19 pneumonia in severe stages can present clinically like bacterial pneumonia) diagnosed? Radiologically? Microbiologically? This is a limitation of population-based study. It is likely not likely useful for ‘pneumonia’ to be included in the analyses as delineation is difficult (unlike COPD, hypertension, diabetes). I would suggest removing this altogether.

: “Pneumonia” in our study indicates the previous history of pneumonia before the confirmation of COVID-19, as we had described in the method section. “Pneumonia” was identified based on diagnostic code in the HIRA database. Please, reconsider this definition.    

4. It is also important to discuss around how these chronic and acute co-morbidities play in making the decision for further escalation such as ICU admission, intubation and ventilation as well advanced resuscitation care.

: As you commented, these comorbidities may play a role in the making the decision for mechanical ventilator, cardiopulmonary resuscitation, or extracorporeal membrane oxygenation. It is possible that patients with underlying diseases have received more intensive care due to concerns that they will be more susceptible to corona infection. We added this notion in the discussion section (highlighted in yellow).

5. Please discuss further Line 218 re: sociocultural factors and quarantine norm of countries more explicitly.

: For example, people in Western countries can be more exposed to close physical contact such as hug and kiss as compared with those in Asian and Middle Eastern countries. In addition, there are significant differences in the degree of the restriction of social contact according to the countries. We added this notion in the discussion section (highlighted in yellow). 

6. For paragraph starting from Line 251, it’s important to emphasize that prescribed medications for Covid-19 is like a confounding bias for more severe patients. Although both uni and multivariable analyses were used, hospitalization may be a useful variable in the logistic regression analyses to reflect severe patients who were hospitalized and likely to be commenced on anti-viral, antibiotics etc.

: We agree your comment. It is evident that prescribed medications for COVID-19 may be considered as a confounding bias for more severe patients. We added this notion in the discussion section (highlighted in yellow). Hospitalization may be a useful indicator in the severity of COVID-19. However, all patients with confirmed COVID19 were hospitalized during the study period in Korea. Thus, we assumed that hospitalization itself do not affect the result of our data.   

7. Also worth mentioning if systemic steroids and/or remdesivir data available due to recent insight on the benefits

: Unfortunately, as #OpenData4COVID19 project ended last August, we are not able to analyze the effect of systemic steroids and/or remdesivir on the mortality and severity of our cohort. In addition, remdesivir had not been available during the study period in Korea.